# Sources of Antifungal Drugs

**DOI:** 10.3390/jof9020171

**Published:** 2023-01-28

**Authors:** Giel Vanreppelen, Jurgen Wuyts, Patrick Van Dijck, Paul Vandecruys

**Affiliations:** KU Leuven Laboratory of Molecular Cell Biology, Institute of Botany and Microbiology, Kasteelpark Arenberg 31, B-3001 Leuven, Belgium

**Keywords:** antifungal, natural products, synthetic compound library, antifungal pipeline

## Abstract

Due to their eukaryotic heritage, the differences between a fungal pathogen’s molecular makeup and its human host are small. Therefore, the discovery and subsequent development of novel antifungal drugs are extremely challenging. Nevertheless, since the 1940s, researchers have successfully uncovered potent candidates from natural or synthetic sources. Analogs and novel formulations of these drugs enhanced the pharmacological parameters and improved overall drug efficiency. These compounds ultimately became the founding members of novel drug classes and were successfully applied in clinical settings, offering valuable and efficient treatment of mycosis for decades. Currently, only five different antifungal drug classes exist, all characterized by a unique mode of action; these are polyenes, pyrimidine analogs, azoles, allylamines, and echinocandins. The latter, being the latest addition to the antifungal armamentarium, was introduced over two decades ago. As a result of this limited arsenal, antifungal resistance development has exponentially increased and, with it, a growing healthcare crisis. In this review, we discuss the original sources of antifungal compounds, either natural or synthetic. Additionally, we summarize the existing drug classes, potential novel candidates in the clinical pipeline, and emerging non-traditional treatment options.

## 1. Introduction

From the foxfire bioluminescent mushrooms to the largest organisms on earth, fungi are diverse, ubiquitous cornerstone members of various ecosystems. Whereas many fungi are beneficial to humans, e.g., for cheese and alcohol production, a number of them also display pathogenic characteristics. Fungal infections pose a continuous global threat to human and animal health, jeopardize entire ecosystems, and place a tremendous burden on food production [1]. Fungi cause a range of infections in humans, from harmless, superficial maladies to life-threatening invasive mycoses. The global acquired immunodeficiency syndrome (AIDS) crisis, the increased use of implants, and the overall improved survival rates of immunocompromised patients have resulted in a steady increase in fungal infections [2,3]. These are associated with relatively high incidence, high mortality rates, and high hospitalisation costs [4]. This is particularly the case for tenacious biofilm-associated infections [5]. Biofilms are complex three-dimensional structures with a typical micro-colony architecture characterized by extensive spatial heterogeneity and an extracellular matrix material associated with increased resistance to host immune factors and antifungals [6]. Due to these characteristics, biofilms frequently allow infections to re-establish after treatment. Therefore, it is no surprise that fungal infections are responsible for 1.4 million deaths on a global scale each year [7].

Treatment options for human fungal infections are currently limited to five different classes of antifungals, of which just three are regularly used as standalone treatments for mycosis. Figure 1 provides a chronological overview of their point of origin and both the discovery and introduction to the market of their most established member. As discussed further in this review, recently, a number of novel classes of antifungals reached the market or are in advanced clinical trials. Here, we will provide a summary of the currently used drug classes.

(1) The polyenes are the oldest antifungal drug class in clinical use today and hallmarked a major clinical advancement in mycosis treatment. Although the class boasts some of the most potent antifungal compounds, they are associated with severe nephrotoxicity [8] and low water solubility [9], which limits their use as antifungal drugs. Amphotericin B (AmB) is the most well-known member of this class. It was purified in 1953 as a secondary metabolite from the secretion broth of the actinobacterium *Streptomyces nosodus* by Elizabeth L. Hazen and Rachel F. Brown [10]. Additional representatives of the polyene drug class currently in use for antifungal therapy are natamycin and nystatin. The latter was unearthed by Hazen and Brown before they discovered AmB. Amphotericin B, and its lipid formulations with enhanced activity, were and are still predominantly used in the clinical context [11]. Polyenes exert their effect by binding sterols, primarily ergosterol, which are crucial components for the stability of the fungal membrane. Although discussion on the exact mode of action of polyenes is ongoing, it is generally accepted that the antifungal effect is caused by sequestration of ergosterol from the fungal cell membrane, by acting as a sterol-sponge [11]. Therefore, fungal exposure to the polyenes causes membrane pore formation, sterol rafts, and reactive oxygen species (ROS) formation, which induces apoptosis. Hence, polyenes are strong fungicidal compounds for most fungal pathogens [12]. Although nephrotoxicity limits the practicality of polyenes in favor of newer antifungals, due to their potent fungicidal effect and low resistance rate, they have remained the last bastion for patients with life-threatening invasive fungal infections for over 70 years.

Initially synthesized as an anti-cancer agent in 1957, flucytosine is the sole member of the (2) pyrimidine analogs antifungal drug class and was introduced in 1971. These pyridine analogs are compounds that mimic the structure of pyrimidine [13]. 5-fluorocytosine (5-FC) is a fluorinated analog of the nucleoside cytosine. The 5-FC drug is administered as a pro-drug and taken up by susceptible fungi through cytosine permease. Upon entry in the cytosol, it is rapidly deaminated by cytosine deaminase to yield 5-fluorouracil (5-FU). This compound exerts its antifungal activity via two distinct mechanisms. First, from 5-FU, fluorodeoxyuridine monophosphate (FdUMP) can be generated through the action of uridine monophosphate pyrophosphorylase. This FdUMP inhibits fungal DNA biosynthesis by acting on thymidylate synthase. Second, 5-FU can also yield fluorouridine triphosphate (FUTP), which can serve as a substitute for uridylic acid. Its integration into fungal RNA inhibits protein synthesis, which disturbs the amino acid pool and amino-acetylation of tRNA [14]. Depending on the fungal pathogen, its effect can either be fungicidal or fungistatic [14]. Although it interferes with fundamental molecular mechanisms, resistance against 5-FC is widespread and has barred the drug from monotherapy treatment. However, it is administered in combination with other drugs, acting synergistically with AmB and with members of the azole drug class [15]. The 5-FC drug is recommended at the early stages of cryptococcal pneumonia and meningoencephalitis. Less frequently, this combination therapy is used for invasive fungal infections in favor of monotherapy [16,17]. However, in regions where access to AmB is limited, 5-FC combined with fluconazole is administered for cryptococcal meningoencephalitis.

Due to the toxic nature of polyene compounds, and the intrinsic tolerance of pathogens to 5-FC, researchers continued to search for novel and safer compounds. This ultimately led to the introduction of the azole drug class in the 1960s, although the antifungal effect of the azole compound benzimidazole had been described in 1944 by Woolley [18,19]. Currently, (3) the azoles are the most widely used antifungals. Initially discovered during a synthetic screening, members of the drug class can be divided into two groups, namely the imidazoles (miconazole and ketoconazole) and the triazoles (fluconazole, itraconazole, voriconazole, posaconazole, and isavuconazole). Due to lower host toxicity, overall better pharmacokinetics, and increased efficacy, the triazoles have replaced the earlier imidazoles starting with fluconazole, which was introduced in 1990 [20]. All azoles act by preventing the biosynthesis of ergosterol, a key membrane component. More specifically, they inhibit cytochrome P450-dependent lanosterol 14-α-demethylase, a crucial step in the ergosterol biosynthesis pathway, by competitively binding its heme group within the substrate pocket [21,22]. Depletion of ergosterol results in the inability of fungal cells to develop mature cell membranes. Additionally, with lanosterol demethylase inhibited, the fungal cell will accumulate toxic methylated sterol intermediates. As a result, the azoles inhibit fungal growth and are in general fungistatic drugs. Although they function through an identical mode of action, there is a distinct difference between different triazoles in efficacy, spectrum of activity, and overall pharmacological parameters. As mentioned before, azoles suffer from rapid resistance development and overall intrinsic tolerance of multiple species, their widespread use in clinical and agricultural settings has exacerbated this trend. Additionally, azoles are subjects of unfavorable drug–drug interactions, which causes significant problems with patients undergoing polypharmacy. If no innovative adaptation or next-generation compounds are found for this chemical drug class, the resistance issue might result in a continuous downfall in applicability.

(4) Allylamine antifungals were discovered by chance in 1974 during efforts to synthesize novel compounds for the treatment of central nervous system disorders. Its most successful member, terbinafine, was approved for topical use in 1992 and subsequently as an oral antifungal in 1996. [23] The drug is commonly used for treating dermatophyte infections. Allylamines work similarly to the azoles by inhibiting the ergosterol pathway. However, they target squalene oxidase, an enzyme upstream in the ergosterol pathway [24]. This causes intracellular squalene accumulation and ergosterol deficiency in the fungal cell membrane. Interestingly, it was reported very recently that researchers purified and identified allylamine compounds from secretion extracts of a *Lysinibacillus* isolate, making this class the only one that has both a synthetic and natural origin [25].

(5) The echinocandins drug class is the newest member in the clinical antifungal armory, introduced in 2001 [26]. They are semisynthetic lipopeptides whose bioactive precursor compounds were purified from fungal secretion extracts [27], and later chemically adjusted to improve solubility and overall pharmacokinetic characteristics. Its first member approved for systemic use, caspofungin, was derived from the fungus *Glarea Lozoyensis*, which secreted pneumocandin B_0_, a lipophilic cyclic peptide. Currently, three echinocandins have been approved for clinical use, namely caspofungin, anidulafungin, and micafungin. More recently a new member, rezafungin from Cidara Therapeutics, was determined a QIDP with fast-track status by the FDA and has obtained an orphan drug designation in the US and EU [28]. Echinocandins inhibit β-(1,3)-glucan synthase, a crucial enzyme in glucan biosynthesis [29], by binding non-competitively to the Fks1 subunit. In general, the fungal cell wall relies heavily on an interlinked polysaccharide skeleton of β-(1,3)-glucan and chitin for structural integrity. Disruption of β-(1,3)-glucan synthesis results, therefore, in compromised cell walls. This causes osmotic instability, lysis of the cell, and ultimately cell death. Echinocandins are restricted to IV use only and are expensive relative to other antifungals. These antifungal drugs are associated with a paradoxical growth pattern, with growth at supra-MIC concentrations, referred to as “the Eagle effect” [30,31]. Depending on the targeted organism, echinocandins are considered as either fungistatic or fungicidal [32].

Several antifungal candidates as representatives of known and potentially novel drug classes are currently in the clinical pipeline and have been extensively reviewed elsewhere. The three compounds belonging to potential novel drug classes will be briefly discussed here, as well as a novel azole formulation [33,34,35].

First, ibrexafungerp (formerly known as SCY-078 and MK-3118) is an antifungal developed by Scynexis. It is currently in phase 3 clinical trials to treat recurrent vulvovaginal candidiasis (RVVC) and invasive fungal infections (NCT05399641) (NCT05178862). Ibrexafungerp received FDA approval on June 1, 2021. The compound employs a similar mechanism of action as the echinocandins; however, as a triterpenoid it has a completely different chemical structure from the echinocandins. Second, olorofim, formerly known as F901318, is a synthetic compound developed by F2G, representing the orotomide drug class [36]. Olorofim inhibits dihydroorotate dehydrogenase [37], a novel target, and an essential part of pyrimidine biosynthesis of molds, such as the *Aspergillus* species. At the time of writing, olorofim is undergoing phase 3 clinical trials for the treatment of invasive aspergillosis (NCT05101187). Third, fosmanogepix, formerly known as APX001, is a synthetic compound discovered in a target-based screening assay [38,39]. It is currently in phase 2 clinical trials for the treatment of non-neutropenic patients with candidemia (NCT03604705) and invasive mold infections (NCT04240886). Fosmanogepix is currently being developed by Pfizer after the company acquired Amplyx Pharmaceuticals. Fosmanogepix is a N-phosphonooxymethylene prodrug that metabolizes into manogepix after administration. It acts by targeting the fungal enzyme GPI-anchored wall transfer protein 1 (Gwt1), which is necessary for the acylation of inositol in the glycosylphosphatidylinositol (GPI) anchor biosynthesis pathway. Since this pathway is vital for proper mannoprotein anchoring to the cell wall, its interruption compromises the structural integrity of the fungal cell wall, which will ultimately lead to growth cessation. Finally, oteseconazole (formerly VT-1161) developed by Mycovia Pharmaceuticals, is a tetrazole, and was granted FDA approval for the treatment of RVVC in April 2022 [40]; it is undergoing phase 3 clinical trials (NCT03562156). The tetrazole has an improved affinity for fungal CYP51 compared to triazoles but an overall lower affinity for heme iron [41]. Tetrazoles might introduce a new, improved trend for the azole drug class.

Despite the availability of several different classes of antifungals, there are still unmet needs that require the development of novel antifungals [33]. Antifungal use is widespread to treat human infections, for crop protection, timber preservation, and treatment of animal infections. This excessive use is leading to a global increase in resistance, primarily to the azole antifungals [42]. Moreover, the emergence of species that are able to acquire multidrug, or even, pan-resistance, such as *Candida auris* [43], complicates treatment options and highlights the need for novel antifungal drug candidates. Given this unmet need, we review potential sources for novel antifungals. Here, we focused on the sources that can be used to screen for novel antifungals and what the implications are when using these sources.

## 2. Natural Products

Historically, natural products have been a rich source of antimicrobials [44]. It all started when Alexander Fleming accidentally discovered penicillin [45]. He observed a mold contamination that visibly inhibited the growth of his staphylococci. Selman Waksman (1944) applied the same principle on a larger scale to screen for antimicrobials produced by *Actinobacteria*. This approach is also referred to as the Waksman platform. As a result, he and his team discovered streptomycin, the first antibiotic active against Gram-negative pathogens that could be used as a drug [46]. Using this same approach, they also identified several antifungal compounds (e.g., candicidin) [47]. Still, it was Hazen and Brown (1951) that discovered nystatin, the first antifungal compound from *Actinobacteria,* that would be developed as an antifungal drug [48].

It comes as no surprise that a large proportion of the antimicrobials that are currently applied are derived from natural products. Consider how millions of years of evolution shaped the continuous arms race between microorganisms, for which the production of antimicrobials offered a competitive edge to survive or even thrive in a certain niche. The further development of these compounds boomed, a success caused by the effectiveness of the Waksman platform, but also due to the inefficiency of synthetic screening campaigns and target-based approaches. Even though hit rates for antifungals from natural product libraries sometimes exceed synthetic screening campaigns up to 200-fold (see Table 1), they tend to be challenging drug candidates. An overall downside of natural products is that they are often large and complex, making de novo synthesis or production of analogs challenging and, consequently, making it harder to establish them as treatment options in the clinical context [49].

It is recognized that microorganisms have a complex life cycle [55]. They often reside in multicellular structures, such as biofilms, and this preferred lifestyle is also reflected in the clinical context. These biofilms are up to 100-fold more resistant to antifungals compared to planktonic cultures and it has been well-estimated that most infections originate from biofilms [56,57,58]. Strikingly, natural product antifungals appear to have higher anti-biofilm activity, compared to synthetic antifungals. Echinocandins and polyenes, both derived from natural products, are associated with strong anti-biofilm activity. In contrast, azoles, allylamines, and pyrimidine analogs are synthetic in origin and exert poor anti-biofilm properties [56,59,60,61]. The reduced efficacy of these antifungals on biofilms is attributed to their sequestration by the extracellular matrix containing β-glucan, which reduces the antifungal concentration able to reach the target cells, resulting in increased tolerance of cells within the biofilm [62,63].

Large-throughput screening campaigns by biotech and pharma companies, but also academia, are well-suited for lead compound discovery [64]. To fill in the gaps, over the last decade, compound libraries containing pure or semi-pure natural products have been composed. A prime example is the compound library of the National Cancer Institute’s Natural Products Branch (NPB). With over 320,000 fractions available for large-scale screening, it holds one of the largest collections of publicly available pre-fractionated natural product libraries [65]. These natural products can be derived from plants, fungi, or bacteria.

**Plant-derived antifungals.** Plants live in timescales that cannot be compared to those of most (micro)organisms. Combined with their sessile lifestyle, plants need defense mechanisms that trigger little to no resistance development, ensuring their usefulness throughout their lifespan. Preferably, these active compounds address various challenges that plants may experience at a given moment, such as predation by rodents or insects, and infection by microorganisms. As a result, plant-derived compounds appear to be mainly toxic with lower specificity, restricting their application scope to anti-cancer or anti-parasitic drugs [66]. Prime examples of aspecific plant-derived natural compounds are curcumin and resveratrol. These compounds have been reported to have antiviral, antibacterial, and antifungal properties (among others) [67,68,69,70]. Despite several hundreds of clinical trials and thousands of publications, these compounds are now regarded as pan-assay interference compounds (PAINs). These molecules are frequent hitters in (phenotypic) screening campaigns and often share structural features that show promiscuous biological activity. Therefore, clinical applications for most of these molecules are unlikely [54]. Notable exceptions here are anti-malaria compounds, such as artemisinin and quinine, which act on malarial mitochondria and purine nucleoside phosphorylase as their specific targets, respectively [71,72]. Interestingly, some anticancer compounds, such as camptothecin and podophyllotoxin, identified in plant extracts, are now assumed to be produced by fungal endosymbionts [73].

**Fungal-derived antifungals.** Remarkably, fungi are among the best producers of antifungals. As with some bacteria, fungi have multidomain non-ribosomal peptide synthetases (NRPS) that can produce peptides without the aid of ribosomes. Although the principle of NRPS is the same, clear differences between fungal and bacterial non-ribosomal peptides exist, such as peptide size distribution and monomer composition. Aside from the final peptide itself, the enzymatic synthesis methodology can strongly differ [74]. Fungal-derived natural products are often unique to a fungal genus or species, since horizontal gene transfer in fungi is rather rare compared to bacteria [75]. Therefore, the isolation of rare fungi is associated with increased chances of isolating novel natural antifungal products. Several medically useful antifungals derived from natural products produced by fungi include the echinocandins and the novel ibrexafungerp [26,47,76]. Both classes of antifungals target the catalytic subunit of β-glucan synthase. These β-glucan synthase inhibitors are the most frequently isolated compounds from fungal extracts but have never been isolated from bacterial sources [50]. The fungi that produce the natural precursors of these drugs all belong to the family of *Trichocomaceae.* They are aggressive colonizers and probably produce antifungals to maximize their potential as saprobes. Generally, they themselves are less susceptible to the antifungals they produce. For example, the echinocandins have strong concentration-dependent fungicidal activity against *Candida* but are only static against *Aspergillus*, a member of the *Trichocomaceae* family. Due to the diversity of the fungal kingdom, these family feuds should be considered when the antifungal development program focuses on different pathogenic lineages, for example, on *Aspergillus* or other members of the *Trichocomaceae family.*

**Bacteria-derived antifungals.** Antifungals derived from bacterial sources, in clinical use today, are all derived from *Actinobacteria*. These aerobic Gram-positive bacteria are highly abundant in soil and marine sediments and constitute one of the largest bacterial phyla [77]. They have a significantly larger genome size compared to other bacteria and a high G/C content. They are self-sustainable, making them easy to isolate and cultivate. Like fungi, they develop a mycelium with spores. During spore formation, the vegetative mycelium undergoes programmed cell death to reallocate nutrients to the spores. To prevent other microbes from using these nutrients, they produce secondary metabolites with antimicrobial activity [7]. Therefore, *Actinobacteria* and especially the *Streptomyces* genus are recognized as specialized producers of secondary metabolites [78,79]. It has been estimated that members of the genus of *Streptomyces* alone could produce up to 100,000 molecules with antimicrobial activity [80]. The potential of these bacteria has been well known for almost a century, resulting in large screening campaigns to exploit the antimicrobial potential of *Actinobacteria*. Cubist Pharmaceuticals, for example, screened over 10^7^ *Actinobacteria* every year and estimated that a novel antibiotic could be discovered at frequencies below 10^−7^ per random *Actinobacteria*. Moreover, they estimated that the global top 10 cm of soil contains 10^25^–10^26^ *Actinobacteria*, leaving plenty of opportunity for further screening. Because the burden of fungal infections was often less recognized in the past, it is unlikely that as many *Actinobacteria* have been screened for antifungal activity as for antibiotic properties.

Despite extensive efforts, so far, the only clinically useful antifungals discovered from *Actinobacteria* were the polyenes. However, due to nephrotoxicity, their implementation is limited [81,82]. Over 200 polyene compounds have been described, mainly from *Streptomyces* [9]. They appear to be the most abundant antifungals produced by *Actinobacteria,* outweighing other antifungals by a factor 20 [47]. A screening by Roemer et al. (2011) confirmed the abundance of polyenes produced by *Actinobacteria.* Moreover, they also concluded that most antifungals produced by *Actinobacteria* appear to lack specific targets, with the majority being ionophores [50]. This resulted in a decreased hit rate for target-specific antifungals derived from *Actinobacteria* (9%), compared to fungi (>50%). One strategy to avoid the rediscovery of polyene antifungals employs the use of a polyene-resistant test strain. However, this resistance is generally associated with a serious fitness cost [83], making it hard to use resistant strains in screening efforts to decrease polyene rediscovery. Fortunately, polyenes can be readily identified in extracts due to their distinct light absorption spectra [84].

Another interesting group of antifungals from *actinobacteria* are the chitin inhibitors, nikkomycins and polyoxins. The latter was derived from *Streptomyces cacaoi* in 1960, while the former was derived from *Streptomyces tendae* in 1976 [85,86]. These peptidyl nucleoside antibiotics are analogs of the substrate UDP-N-acetylglucosamine and, therefore, act as competitive inhibitors of chitin synthase. Since chitin is a crucial component of a stable fungal cell wall and is absent in mammalian cells, it is generally considered a promising drug target [87]. Polyoxin D showed in vitro activity against *Coccidioides immitis*, *Cryptococcus neoformans*, and *C. albicans*, but failed to remain consistent during in vivo murine assessments [88,89,90]. The compound nikkomycin Z showed potent activity against some infections, such as coccidioidomycosis, but only displayed moderate activity against *Histoplasma capsulatum*, *C. albicans*, and *C. neoformans*. Furthermore, filamentous fungi and non-*albicans Candida* species were practically resistant. It does, however, work synergistically with glucan synthesis inhibitors and triazoles [91,92,93,94,95]. It underwent clinical trials in the 1990s, but the bankruptcy of the sponsoring pharmaceutical companies resulted in the termination of ongoing trials. Stranded as a research topic, the project was continued by the University of Arizona, which reactivated the clinical studies [93,96,97,98,99,100].

Other notable antifungals that have been discovered more recently from actinobacterial sources include bafilomycins, neomaclafungins, astolides, caniferolides, and azalomycin F [101,102,103,104,105]. However, several of these compounds also inhibit the growth of mammalian cells and bacteria, thereby diminishing their potential for development as medically useful antifungals.

*Streptomyces* are, historically, the most successful bacterial genus in terms of antifungal drug discovery thanks to the polyenes which have become a cornerstone in mycosis treatment. However, other genera also stood out due to their remarkable antifungal activity. *Bacillus* and *Pseudomonas* species have numerous records in the literature reporting their antifungal potency. *Pseudomonas aeruginosa* is a prominent opportunistic pathogen that displays an antagonistic relationship with fungal pathogens during co-infection. It secretes an array of metabolites to overcome fungal competitors during infection; as such, these metabolites are often characterized as essential virulence factors of the pathogen. These include lactones, alkyl quinolones, rhamnolipids, phenazines, and siderophores such as pyrrolnitrin. Most act as crucial quorum sensing molecules, iron scavengers, and overall virulence factors [106,107,108,109,110,111,112]. Although they exhibit strong antifungal activity, often these metabolites suffer from host toxicity, making further drug development challenging. *Bacillus* species, especially its most known member *bacillus subtilis*, have long been known for their biocontrol properties, tackling diseases caused by fungal phytopathogens. Their antifungal activity has been attributed to a multitude of compounds including but not limited to lipopeptides (surfactins, iturins, fengycins), polyketides (bacillaene, macrolactin), enzymes, such as chitinases, and volatile compounds, such as pyrazine [113,114,115,116,117,118,119,120,121,122,123,124,125,126,127,128,129]. Although an increasing number of antifungal agents have been identified and purified from both *bacillus* and *pseudomonas* species, none have been able to make it through drug development for clinical adaptation.

**Microbial dark matter.** Most bacteria and, to a lesser extent, fungi cannot be cultivated in standard laboratory conditions [130]. In natural ecosystems, this so-called “microbial dark matter” makes up roughly 99% of the microorganisms and comprises a diverse set of microorganisms. Undoubtedly, unknown natural compounds with antimicrobial properties stay hidden as this vast potential remains unmined. Soil-derived microorganisms can roughly be divided into three classes.

The first class, the cultured minority, comprises less than 1% of the total amount of microorganisms. Almost all bacteria in this group belong to only four phyla, namely *Actinobacteria*, *Proteobacteria, Bacteroidetes,* and *Firmicutes*. All microbial-derived antimicrobials used today come from this group, but the low-hanging fruits of this group have been picked [131].

The second class is the in situ cultivable group. These microorganisms cannot be immediately cultivated in a laboratory environment because growth factors, such as siderophores, are missing [131]. Cultivating these microorganisms requires more advanced methods, such as, for example, the isolation chip (iChip) developed by Nichols et al. (2010). This device holds miniaturized microbial growth chambers where single cells are confined and separated from the environment by a semi-permeable membrane [132]. This protects slow-growing species from aggressive colonizers that often dominate samples cultivated in the lab. Additionally, growth factors essential for germination or growth produced by other microorganisms or present in the soil can permeate through the membranes, making proliferation possible, which results in pure cultures of potentially novel microorganisms. Although in situ cultivation can be used to isolate a larger proportion of uncultivable microorganisms from soil samples, this approach rarely results in the isolation of microorganisms from uncultured phyla. Instead, the in situ cultivated microorganisms are usually rare or less cultured members of *Actinobacteria*, *Proteobacteria,* and *Firmicutes* [132]. Because these organisms are closely related to microbes that have already been extensively screened, a large proportion will likely produce the same or highly similar antimicrobials. Nevertheless, rare isolates can yield novel antimicrobials. This approach has already proven its success with the discovery of the promising antibiotic teixobactin [133]. Still, it remains to be seen whether antifungals discovered using this platform find their way to the clinic.

The third and final class are microorganisms that cannot be readily cultured in standard laboratory conditions, even when in situ cultivation devices, such as the iChip, are used. In terrestrial habitats, these microorganisms belong to phyla lacking cultured representatives, such as *Acidobacteria*, *Chloroflexi,* and *Planctomycete*. Cultivation is hard, if not impossible, for this group. It has been suggested that some of them are intrinsically slow growers and that cultivation is only possible after growing them for several months in the lab while retaining the correct conditions [134,135]. It remains unclear why some of these uncultured microorganisms are so abundant in the soil [130]. Probably, some necessary factors are still lacking to cultivate these microorganisms in a lab environment. Moreover, it is unknown whether these microorganisms can produce antimicrobials since they generally have relatively small genome sizes, ranging from only 0.148 Mb to 2.4 Mb [136]. It has been estimated that below a genome size of 3 Mb, polyketide synthase (PKS) and non-ribosomal peptide synthase (NRPS) genes are absent or rare [137]. As per the current literature, since these genes are critical components of secondary metabolite pathways, it is unlikely that they are abundant producers of secondary metabolites [79]. Still, it can also not be excluded that species with a small genome size encode antimicrobial molecules that are not encoded by NRPS or PKS operons. In contrast, as mentioned before, *Streptomyces coelicolor* has a genome size of 7.6 Mb, of which 5–10% of its genomic sequence is dedicated to secondary metabolite production [78,138]. Only a limited number of uncultured microorganisms have been whole-genome sequenced so far. Thus, all we need is the discovery of a novel genus from the microbial dark matter with a similar coding potential to *Streptomyces*, enabling us to unlock another era of highly successful natural product discovery.

Although beneficial, strain cultivation is not necessary for antimicrobial discovery. A recent study discovered the malacidin class of antibiotics using a culture-independent approach [139]. A polymerase chain reaction-based approach was used to amplify calcium-dependent antibiotic gene clusters directly from soil samples. After heterologous expression in the model host *Streptomyces albus*, secretion extracts were screened for antibiotic activity resulting in the isolation and purification of malacidins.

## 3. Synthetic Compounds

Whereas most antibiotics are of natural origin, the most frequently used antifungals, the azoles, are of synthetic origin. The reason for this may be the relatively late interest in antifungal drug discovery. After several decades of steady mortality rates due to candidiasis, in 1970, mortality rates increased substantially. This rise can be attributed to the use of immunosuppressive therapies, the increase in immunodeficient patients, such as those suffering from human immunodeficiency virus (HIV) infections, the increased use of antibacterial agents with a broad spectrum, and the frequent use of indwelling intravenous devices. Only in the 1980s were invasive mycoses recognized as a health threat [140]. Consequently, when large-scale screening platforms emerged, the focus resided on bacteria rather than fungi. So far, only one class of antifungals approved for standalone systemic use, the azoles, are derived from synthetic compound libraries [141,142].

Synthetic compounds are the result of available techniques and a chemist’s imagination. Consequently, they occupy a more limited chemical space than natural products [66]. Since the outcomes of these screening efforts are restricted by the envisioned goal and pharmaceutical and chemical parameters of the included compounds, compound libraries are generally biased [143,144]. One way to resolve this is by using a synthetic compound library that is comprised of a diverse set of compounds. For example, the Community for Open Antimicrobial Drug Discovery (CO-ADD) has composed a library of chemical compounds from academic sources and is continuously using crowdsourcing to increase its library size [145]. A proof-of-concept screening resulted in 20–30 times higher hit rates for bacteria (compared to commercially available libraries) and a hit rate of 0.98% for fungi [146]. This library will also be used to screen against the fungal targets *C. albicans* and *C. neoformans*.

Imidazole and triazole pharmacophores are relatively abundant in these libraries and are estimated to constitute around 15% of the hits when screening these libraries against the opportunistic fungal pathogen *Candida albicans* [50]. Compounds that enter these synthetic libraries need to pass through filters to make them a “good drug” later in the development process. One such rule is the Lipinski rule of five (RO5) which states that molecules should have a limited size and a lipophilic nature to ensure a good oral bioavailability [147]. Therefore, most compound libraries are biased toward compounds that follow these rules but do not necessarily have good antifungal properties. These rules were defined by comparing the properties of compounds that made it through the first phase of clinical trials. However, recently it was disproven that molecular weight can be used to predict oral bioavailability [148]. An additional advantage of these low molecular weight (<500 Dalton) compounds was their relative ease of synthesis. Therefore, when RO5 was established in 1997, synthetic compound libraries contained relatively smaller molecules resulting in a bias towards smaller molecules that were used as drugs. The molecular weight of approved drugs has been steadily increasing over the past years, and it has been estimated that good absorption drops sharply above 975 Dalton [149], almost twice the size originally described in RO5. It is expected that higher molecular weight molecules will be added to synthetic compound libraries in the future, and this concomitant increase in complexity could also yield higher hit rates against fungi.

Synthetic compound libraries are often used for target-based drug discovery, while natural products are more often screened in whole-cell assays. Unfortunately, target-based antifungal drug discovery faces identical issues to antibiotic-based drug discovery, and has so far failed to yield a clinically applied antimicrobial [50,134,150]. One study exemplifies the difficulties that an in vitro target-based screening can encounter during translation to in vivo viability screens [51]. During this study, the activator–mediator interaction responsible for *Candida glabrata* azole resistance (Pdr1 activation domain and the Gal11A KIX domain) was targeted. In this screen, small molecules that could inhibit this interaction would re-sensitize drug-resistant *C. glabrata* to azole antifungals. A dozen synthetic compound libraries were screened, totaling over 143,000 compounds. This resulted in 352 potential inhibitors in an in vitro screening. Due to the presence of the fungal cell wall, a large proportion of these active compounds lacked the ability to penetrate the fungal cell wall envelope. Only five compounds showed activity on live cells, with iKIX1 as the most promising lead, corresponding to less than 2% of all in vitro hits.

Despite the challenges associated with the screens using synthetic compound libraries, they can still be a very successful approach, as shown by the recent discovery of F901318 (olorofim) [37]. In this study, the F2G company screened 340,000 compounds against the airborne pathogenic mold *Aspergillus fumigatus* and discovered a novel chemical series with potent activity against *Aspergillus* species, but with no activity against *C. albicans.* This might explain why these compounds went unnoticed in previous screening campaigns, because *Candida* was typically the target pathogen. This indicates that using a panel of different fungi as targets can reveal novel compounds with a novel mode of action.

## 4. Non-Traditional Antifungal Options

Aside from the small-molecule drug treatment for mycosis, other options, so-called non-traditional antifungals are being assessed by research and, thus, have not been employed in the clinical context. These so-called “non-traditional antifungals” include peptides, antibodies, vaccines, immunomodulating compounds, mycophages, and virulence factor inhibitors [33,151,152].

Peptides are promising antifungal alternatives [153]. Antifungal peptides (AFPs) originate from a natural source or can be synthesized. In December 2022, the antimicrobial peptide database (APD3) contained 1,277 AFPs [154]. Notable examples of antifungal peptides include LL-37 and histatin 5 [155,156]. They generally exert their antifungal effect by disrupting the cell and/or mitochondrial membrane. The AFPs can form an alpha helix structure, beta-hairpin, a cysteine residue sheet, or a mixture of these conformations upon interaction with membranes. At present, the majority of AFPs were discovered by testing their bioactivity during in vitro screening efforts. Bednarska et al. (2016) developed a platform to target bacterial proteins by cross-aggregation [157]. This platform could be extended to fungi as well. However, several hurdles need to be overcome before AFPs can be successfully used as drugs. They are unstable, have a low half-life, and low bioavailability [158]. Additionally, posttranslational modifications play a crucial role in their biosynthesis and antifungal properties, and in silico predictions of these modifications and overall structure-activity relationship (SAR) remain challenging. However, with the advancements in up-and-coming in silico prediction techniques, the AFP field might see breakthrough advances within the coming decade.

The host’s innate and adaptive immune response and its effector molecules, antibodies, control fungal infectious agents by preventing entry, restricting replication, and modulating the immune response to clear the infection. This combat tool can be exploited by creating antibodies can mark and combat microbes with high specificity. This promising strategy has the potential to alleviate the resistance crisis and can be vital for immunocompromised patients who need immune support [159]. Mycograb is such an example. It is a human recombinant monoclonal antibody that binds heat shock factor 90 (Hsp90) specifically at the site that enables the conformational change upon ATP binding, thereby inhibiting its function [160]. It showed promising synergy with the established antifungal drugs fluconazole, caspofungin, and amphotericin B, but was unable to demonstrate standalone activity in vitro against *C. albicans* [161,162]. It did, however, demonstrate therapeutic efficacy in an invasive candidiasis trial [163]. Unfortunately, no approval for clinical use was obtained from the European Medicines Agency, and this trial was terminated because of quality concerns due to autoaggregation of the antibody. Despite novel formulations, therapeutic application remained out of reach for Mycograb, resulting in the discontinuation of the development of this antibody. Nevertheless, Mycograb provided a vital proof of concept. Antibodies targeting another part of the cell wall or even receptors, such as GPCRs (if accessible), could be valuable sources of antifungals in the future.

Vaccine development to prevent fungal infections poses an interesting alternative because the at-risk patient groups that would benefit from a vaccine are well recognized [164]. Five vaccine categories exist for human fungal diseases, although there is still a long way to go before commercial application. They are listed as follows with a representative example [165]: (1) Live-attenuated strains, such as the *C. neorformans* strain lacking the sterylglucosidase enzyme [166]; (2) killed fungal strain, such as the vaccine for coccidioidomycosis of formalin-killed spherules [167]; (3) fungal extracts, such as glucan particles consisting of *Cryptococcus* alkaline extractions [168]; (4) DNA or RNA vaccines delivering nucleic acid-encoded antigens, such as the DNA vaccine against *Penicillium marneffei* using the cell wall antigen Mp1p [169]; (5) purified fungal associated macromolecules, proteins, peptides, carbohydrates, and lipids. For example, the previous mentioned NDV-3A vaccine utilizes a recombinant agglutinin-like sequence 3 protein from *C. albicans* [168].

Despite the availability of both bacterial and viral vaccines, only a few fungal vaccines have been enrolled in clinical trials. Even though encouraging results have been obtained, until now, they have failed to deliver [165,170]. A promising vaccine so far is the NDV-3A vaccine to treat RVVC. It has shown promising results in a phase 2 trial and has also shown a protective effect in mice against *S. aureus* infection [171,172,173]. So far, vaccine development to combat *C. albicans* has received much attention but developing a vaccine for this fungal pathogen is hard because of three key reasons [170]. First, *C. albicans* is well known for its phenotypic plasticity. It can exist in varying phenotypes, including in yeast form, as pseudohyphae and hyphae, while concomitantly being present as white, grey, or opaque cells, making the selection of epitopes hard [174]. Second, *C. albicans* has evolved as an obligate commensal. Consequently, the immune system naturally tolerates *C. albicans* when it is non-pathogenic. Therefore, training the immune system to act against *C. albicans* feels contradictory. Third, candidiasis generally occurs in immunocompromised patients and, thus, mounting an immune response in patients with an out-of-commission immune system is difficult [170].

Immunomodulatory compounds are another interesting avenue to treat fungal infections. This strategy has been propagated due to advances and successes in immuno-oncology [33]. One example of such a strategy is the targeting of the JUN terminal kinase 1 (JNK1), a negative regulator of the host’s innate immune response. Inhibitors of JNK have demonstrated efficacious antifungal resilience in mice [175]. However, similar to hurdles in vaccine development, patients with mycosis often have an attenuated immune system; therefore, developing immunomodulatory compounds is challenging. Moreover, these types of compounds should not aggravate the immunopathology of the patient or interfere with the effects of other immunomodulatory drugs that are already used to treat an underlying disease. Finally, due to different immunologic conditions, these compounds will likely be useful for some, but not for others.

Although not as prevalent as bacteriophages, mycophages might represent a treatment option for fungal infections [176]. Despite not being studied as much as bacteriophages, mycophages have been studied predominantly in *Aspergillus* [177], but have also been reported in *Candida* [178]. It has been shown that some mycophages can be transferred from *Aspergillus* to *Saccharomyces* [179], demonstrating that mycophages can have a broad host range. Infection of *Aspergillus* with mycophages has been shown to result in a suppression of mycotoxin production, a decreased growth rate and a reduced spore formation [177]. However, viral infection of fungi can sometimes also lead to hypervirulent fungal strains. It is important to note that bacteriophages face tremendous challenges as a treatment option, and it is likely that mycophages will face similar if not more stringent challenges [180]. For example, bacteria can develop resistance to phages fast, phages have a limited host-range (often specific to only a subset of strains), and their large size limits tissue distribution. To resolve the limited host-range and resistance issues, mixed formulations of different phages are used, sometimes containing up to 10 different phages, which increases the complexity and cost during production and storage.

Finally, virulence factor inhibitors have gained attention recently. The major advantage of virulence factor inhibitors is that they do not directly kill cells, nor do they inhibit the growth of cells. Instead, as the name implies, they inhibit their virulence capabilities and, therefore, do not exert as much evolutionary pressure to develop resistance as a traditional antifungal. However, this has recently been refuted for bacteria, so it is expected that this could also be the case for fungi [180]. One of the most attractive targets for antifungal virulence factor inhibitors is the yeast-to-hyphae switch, since it is critical for virulence. Antisense therapy using antisense oligonucleotides (ASOs), blocking the expressions of genes necessary for virulence, may be a promising avenue. The ASOs complementarily bind to their target mRNA by Watson–Crick base pairing. As such, they can restore, reduce, or inhibit the target protein expression. This approach has been investigated in other medical fields and has gained FDA approval to treat several afflictions. Unsurprisingly, antisense therapy is being studied as a novel laboratory tool within the fungal field and as a potential innovative antifungal strategy for the treatment of mycoses [181,182,183]. Since the most common fungal pathogens, *Candida, Aspergillus*, and *Cryptococcus* species, have been studied for decades, their virulence traits are well documented. Antisense therapy could, therefore, benefit greatly from this fundamental knowledge. Recently, Araùjo and coworkers (2019, 2022) have developed an ASO targeting the *EFG1* mRNA, which encodes for a central transcriptional regulator of morphogenesis. They reported promising in vitro results as well as validation in a *galleria mellonella* assay [184,185]. It should be noted that currently used antifungals, such as AmB, also block this transition at sub-MIC concentrations [186,187]. Moreover, during drug development, a novel antifungal should have at least as good (non-inferiority study) or better (superiority study) activity than the currently used drugs [188]. *Candida* infections are still associated with mortality rates of roughly 40%, indicating a clear need for novel antifungals. The echinocandins and polyenes are fungicidal and associated with potent anti-biofilm activity. Additionally, their resistance development frequency is relatively low or absent. Infections in non-immunocompromised patients are controlled by antimicrobials and the few remaining fungi are cleared by the immune system. In contrast, systemic fungal infections, such as candidiasis, rarely occur in patients with healthy immune systems. Consequently, the immune system is inadequate to clear the infection. Because virulence factor inhibitors by definition cannot clear the infection, they will probably be useless as curatives in fungal infections. Hence, their application might be as an adjuvant to antifungal drugs during treatment rather than as a standalone therapy. However, anti-virulence compounds could be useful in non-lethal and chronic infections. Examples include RVVC infections and infections of the skin and nails. A virulence factor inhibitor could demonstrate value in these types of infections as a prophylactic where the number of episodes can be used as an endpoint.

Most non-traditional antifungals must overcome additional hurdles. For example, a MIC determination is not always available or even impossible to determine. This requires another way to determine an efficacious concentration. This is especially problematic for immunomodulatory compounds that do not directly exert their effect on fungal cells. Additionally, defining success in vivo during clinical trials will also require novel endpoints for most drugs [180,188].

## 5. Conclusions

The increase in antifungal resistance, the emergence of multidrug-resistant species, and the grim mortality rates of fungal infections all amount to the continuous need for novel antifungal compounds. Several sources from natural and synthetic origins were successfully mined in the past and currently still deliver valuable lead compounds for drug development. However, novel drug discovery proves challenging due to the frequent re-discovery of known antifungals and the identification of molecules that already belong to the scarce antifungal drug classes currently in use. To expand the repertoire of antifungal agents, one could look at underexplored niches. Since most screening efforts took broad, rudimentary soil samples, microorganisms that are restricted to specific niches, such as the rhizosphere or nests of social insects may prove to be a rich source of bioactive compounds with antifungal properties. Moreover, the creation of large natural product-based libraries and a transition of synthetic libraries towards a more diversity-oriented composition will allow the exploitation of a larger fraction of the chemical space, and eventually, expand the antifungal toolbox. Aside from expanding the chemical vision, tailoring screening goals to specific pathogenic species, rather than to fungi altogether, will allow the discovery of family-specific bioactive agents.

Although the field has experienced a standstill of more than two decades since the introduction of the last distinct antifungal drug class, there is hope on the horizon as promising candidates from both synthetic and natural origins are currently in the final phases of development. Like their predecessors, these compounds might become the progenitor of a new class, alleviating the resistance crisis by expanding the therapeutic options. Furthermore, as in oncological, bacterial, and viral research, non-traditional antifungal research efforts, such as vaccines, antibodies, anti-virulence factors, immunomodulatory compounds, and mycophages could revolutionize the field and the way patients are treated. Finally, there exists extensive fundamental knowledge of fungal pathogens’ essential genes and their corresponding proteins. Therefore, interesting antifungal drug targets have long been identified, although identification of compounds that efficiently disrupt said targets without causing harm to the host is challenging. Anti-sense therapy with tailored ASOs could overcome this issue and become a new source of man-made antifungal compounds.

## Figures and Tables

**Figure 1 jof-09-00171-f001:**
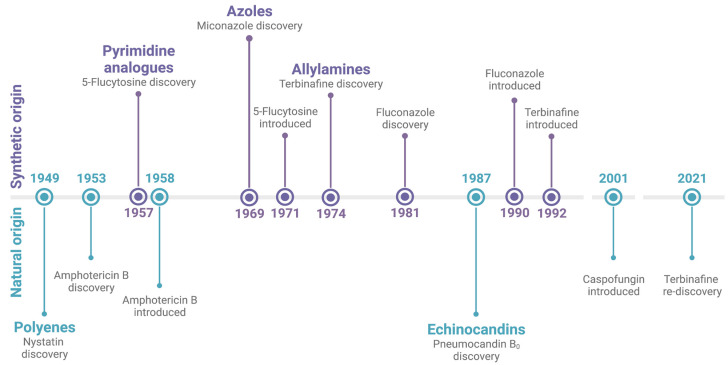
Timeline of the antifungal drug classes. The initial point of discovery of the class itself and both the discovery and introduction to the market of their most established member are depicted. The drug classes and their respective compounds are divided based on their origin, either synthetic (top) or natural (bottom). Created with BioRender.com.

**Table 1 jof-09-00171-t001:** An overview of what to expect when setting up a screening to find (novel) antifungals.

	Natural Products		Synthetic Compounds
	Fungi-Derived	Bacteria-Derived	
**Most common hits**	B-glucan synthase inhibitors (12% of hits) [50]	Polyenes (95% of hits) [47]	Azoles (15% of hits) [50]
**Hit rates (%)**	~20% [50]	~9% [50]	~0.1–2% [51,52,53]
**Common PAINs**	Mycotoxins [50]	Ionophores [50]	Rhodanines [54]

## Data Availability

Not applicable.

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
