# Peer review of "Sources of Antifungal Drugs"

_jof, 2023, doi:10.3390/jof9020171_

Round 1

Reviewer 1 Report

Comments to the author

This is an interesting review, summarizing all possible molecules that can be used in fungal infections.

I have some aspects that you need to consider:

In the final section of virulence factor inhibitors, you can explore also another possible strategy using the antisense oligonucleotides (ASOs). ASOs can be used to block the expression of some genes involved in the virulence factors. Recently, Araújo et al. developed an ASO to target the EFG1 gene that is important in the transition from yeast to hyphae in Candida albicans species. Please see the studies below:

2019 | Araújo, D.; Azevedo, N.M.; Barbosa, A.; Almeida, C.; Rodrigues, M.E.; Henriques, M.; Silva, S. Application of 2’-OMethylRNA antisense oligomer to control Candida albicans EFG1 virulence determinant. Molecular Therapy – Nucleic Acids, 18, 508-517. DOI: 10.1016/j.omtn.2019.09.016. (Q1; SJR: 2.21)

2022 | Araújo, D.; Mil-Homens, D.; Henriques, M.; Silva, S. Anti-EFG1 2’-OMethylRNA oligomer inhibits Candida albicans filamentation and attenuates the candidiasis in Galleria mellonella. Molecular Therapy – Nucleic Acids, 27, 8, 517-523. DOI: 10.1016/j.omtn.2021.12.018 (Q1; SJR: 2.21)

Minor comments:

Line 76: “deanimated”, Do you mean denominated?

Line 164: The second novel drug, called as olorofim is already in clinical trial as the other two?

Line 217: Please use another formation of the table, maybe with horizontal lines to be easier to read.

Line 398: “occupy”. I think that you can replace by “take”.

Line 432: “has thus far…”. Maybe you can use “has so far..”

Line 487: A full stop is missing.

Line 522-531: This paragraph is repeated.

Author Response

This is an interesting review, summarizing all possible molecules that can be used in fungal infections.

I have some aspects that you need to consider:

In the final section of virulence factor inhibitors, you can explore also another possible strategy using the antisense oligonucleotides (ASOs). ASOs can be used to block the expression of some genes involved in the virulence factors. Recently, Araújo et al. developed an ASO to target the EFG1 gene that is important in the transition from yeast to hyphae in Candida albicans species. Please see the studies below:

2019 | Araújo, D.; Azevedo, N.M.; Barbosa, A.; Almeida, C.; Rodrigues, M.E.; Henriques, M.; Silva, S. Application of 2’-OMethylRNA antisense oligomer to control Candida albicans EFG1 virulence determinant. Molecular Therapy – Nucleic Acids, 18, 508-517. DOI: 10.1016/j.omtn.2019.09.016. (Q1; SJR: 2.21)

2022 | Araújo, D.; Mil-Homens, D.; Henriques, M.; Silva, S. Anti-EFG1 2’-OMethylRNA oligomer inhibits Candida albicans filamentation and attenuates the candidiasis in Galleria mellonellaMolecular Therapy – Nucleic Acids, 27, 8, 517-523. DOI: 10.1016/j.omtn.2021.12.018 (Q1; SJR: 2.21)

We have now included antisense therapy in the antivirulence paragraph with the references from Araùjo et al., and wrote lines 577-585, and additional lines 634-640 in the conclusions.

Minor comments:

Line 76: “deanimated”, Do you mean denominated?

We mean deaminated, as it was hinting on the deaminase enzyme activity.

Line 164: The second novel drug, called as olorofim is already in clinical trial as the other two?

We now added clinical trial information on olorofim and also added identification numbers for clinical trials to ibrexafungerp, olorofim and oteseconazole to be more consistent with the information provided with fosmanogepix

Line 217: Please use another formation of the table, maybe with horizontal lines to be easier to read.

The table has been improved

Line 398: “occupy”. I think that you can replace by “take”.

done

Line 432: “has thus far…”. Maybe you can use “has so far..”

done

Line 487: A full stop is missing.

A full stop was added.

Line 522-531: This paragraph is repeated.

corrected

Reviewer 2 Report

The Authors have summarized the currently available drug classes, their mode of action, the most important side effects and spectrum. Regarding that these traditional antifungals clinical efficacy is frequently suboptimal new therapeutic approaches are mandatory to fight against invasive fungal infections. Original sources of antifungals and non-traditional treatment options were also discussed.

Minor comments

Page 3, lines from 136 to 139: β-(1,3)-glucan is the correct.

Page 4, lines: 160-161: recurrent vulvovaginal candidiasis was abbreviated. Later the abbreviation was not used.

Page 11. Lines 496-506 are same with lines 522-531.

Page 12: Two paragraphs were started with “Finally”.

Regarding the origin of nikkomycin Z, it can be mention in this review.

Author Response

The Authors have summarized the currently available drug classes, their mode of action, the most important side effects and spectrum. Regarding that these traditional antifungals clinical efficacy is frequently suboptimal new therapeutic approaches are mandatory to fight against invasive fungal infections. Original sources of antifungals and non-traditional treatment options were also discussed.

Minor comments

Page 3, lines from 136 to 139: β-(1,3)-glucan is the correct.

done

Page 4, lines: 160-161: recurrent vulvovaginal candidiasis was abbreviated. Later the abbreviation was not used.

done

Page 11. Lines 496-506 are same with lines 522-531.

corrected

Page 12: Two paragraphs were started with “Finally”.

adjusted

Regarding the origin of nikkomycin Z, it can be mention in this review.

We wrote a paragraph on nikkomycin Z under bacterial derived antifungals. Chitin synthase inhibitors were indeed a good suggestion to mention and have therefore been included.

Reviewer 3 Report

The review conducted by Vanreppelen and colleagues appropriately covers recent advances in antifungals. The authors' ability to synthesize was evidenced by the number of publications used. I only suggest that the authors describe the acronyms AIDS (line 33) and HIV (line 391) and format Table 1 (lines are missing).

Author Response

The review conducted by Vanreppelen and colleagues appropriately covers recent advances in antifungals. The authors' ability to synthesize was evidenced by the number of publications used. I only suggest that the authors describe the acronyms AIDS (line 33) and HIV (line 391) and format Table 1 (lines are missing).

Done

Reviewer 4 Report

Interesting review on antifungals, especially antifungal drugs,  focused on sources of their origin. This is a novel approach, definitely worth noting.

The ms. is well-written and should be accepted for publication in JoF after minor revision, including some language polishing. My only concern is in my opinion insufficient addressing of antifungals of bacterial origin. Although the authors are perfectly right indicating polyene macrolides as the most important antifungal antibiotics produced by Actinobacteria, much more  antifungals of bacterial origin are known. These include, for example, peptide-nucleoside antibiotics inhibiting chitin synthase (nikkomycin, polyoxin) and several antifungals of amino acid or oligopeptide structure produced by Bacilli. These compounds should be at least mentioned in this review, to show that not only the products of polyketide biosynthetic pathway but also several compounds derived from other pathways of secondary biosynthesis have been found in procaryotic microorganisms.

Author Response

Interesting review on antifungals, especially antifungal drugs,  focused on sources of their origin. This is a novel approach, definitely worth noting.

The ms. is well-written and should be accepted for publication in JoF after minor revision, including some language polishing. My only concern is in my opinion insufficient addressing of antifungals of bacterial origin. Although the authors are perfectly right indicating polyene macrolides as the most important antifungal antibiotics produced by Actinobacteria, much more  antifungals of bacterial origin are known. These include, for example, peptide-nucleoside antibiotics inhibiting chitin synthase (nikkomycin, polyoxin) and several antifungals of amino acid or oligopeptide structure produced by Bacilli. These compounds should be at least mentioned in this review, to show that not only the products of polyketide biosynthetic pathway but also several compounds derived from other pathways of secondary biosynthesis have been found in procaryotic microorganisms.

We wrote a paragraph on niccomycin Z and polyoxin (lines 320-339). Chitin synthase inhibitors were indeed worthy to be mentioned.

We now included a paragraph (lines 342-361) touching the subject of antifungal compounds derived from both Pseudomonas and Bacillus species, the reviewer was just in stating that oligopeptides from bacilli also make for an interesting antifungal group, although, none of them has made it through clinical trials. however, literature suggest that aside from the actinomycetes successful genus of streptomyces both pseudomonas and bacillus have potent antifungal properties. Therefore, both genera have been mentioned in a concise manner.